# Preparation of Biocompatible Manganese Selenium-Based Nanoparticles with Antioxidant and Catalytic Functions

**DOI:** 10.3390/molecules28114498

**Published:** 2023-06-01

**Authors:** Yang Yu, Peng Fan, Jinfeng Li, Shige Wang

**Affiliations:** School of Materials and Chemistry, University of Shanghai for Science and Technology, No. 516 Jungong Road, Shanghai 200093, China

**Keywords:** manganese dioxide, antioxidant, catalytic, tumor microenvironment

## Abstract

The specificity of the tumor microenvironment (TME) severely limits the effectiveness of tumor treatment. In this study, we prepared a composite nanoparticle of manganese dioxide and selenite by a one-step redox method, and their stability under physiological conditions was improved with a bovine serum protein modification to obtain MnO_2_/Se-BSA nanoparticles (SMB NPs). In the SMB NPs, manganese dioxide and selenite endowed the SMB NPs with acid-responsive and catalytic, and antioxidant properties, respectively. The weak acid response, catalytic activity, and antioxidant properties of composite nanoparticles were verified experimentally. Moreover, in an in vitro hemolysis assay, different concentrations of nanoparticles were incubated with mouse erythrocytes, and the hemolysis ratio was less than 5%. In the cell safety assay, the cell survival ratio was as high as 95.97% after the co-culture with L929 cells at different concentrations for 24 h. In addition, the good biosafety of composite nanoparticles was verified at the animal level. Thus, this study helps to design high-performance and comprehensive therapeutic reagents that are responsive to the hypoxia, weak acidity, hydrogen peroxide overexpression nature of TME and overcome the limitations of TME.

## 1. Introduction

In recent years, with the increase in the incidence ratio of cancer, the number of deaths worldwide is also increasing. Acidic pH, hypoxia, excess hydrogen peroxide (H_2_O_2_), and high levels of glutathione (GSH) are specific features of the tumor microenvironment (TME), which is closely associated with tumorigenesis, invasion, and metastasis, and has become a major obstacle in current cancer treatment [1,2,3]. Therefore, the regulation of TME plays a crucial role in the process of tumor treatment.

Manganese dioxide (MnO_2_) has been extensively studied for bio-sensing and bio-imaging due to its high specific surface area and robust pH-responsiveness properties [4,5,6,7,8,9]. MnO_2_ can be used as a drug carrier which can be rapidly decomposed in acidic and reducing environments to achieve drug release. The TME-dependent drug release system can make the therapeutic results specific for tumor tissues without harming normal tissues. For example, Dong et al. synthesized a novel near-infrared (NIR) absorbing photosensitizer SAB by introducing thiophene and iodine substituents on the aza-BODIPY core. Then, MnO_2_ NPs with a large surface area and three-dimensional hydrangea structure were used as carriers loaded with the chemotherapeutic drugs adriamycin (DOX) and SAB, and later modified with amphiphilic polyvinylpyrrolidone to finally obtain MDSP NPs with good physiological stability [10]. MDSP NPs with significant photothermal effects and efficient oxygen autogenesis demonstrated excellent chemokinetic/photokinetic/photothermal synergistic anti-tumor ability [10]. In addition, MnO_2_ was shown to react with the overexpressed H_2_O_2_ in cancer cells by continuously generating O_2_ in situ [11,12]. The O_2_ generated by MnO_2_ can be used to enhance the therapeutic performance of photodynamic therapy (PDT), radiotherapy, and sonodynamic therapy. For example, Pu et al. synthesized hybrid semiconductor polymer nanoparticles and manganese dioxide (MnO_2_) nanosheets as cores and coatings, respectively. These nanoparticles enhanced the tumor’s PDT by generating O_2_ in the tumor microenvironment [13].

It is well known that under normal conditions, intracellular ROS are maintained at constantly low levels and can act as cell signaling pathways, promote mitosis, and facilitate cell proliferation and survival [14,15]. However, excessive ROS tends to cause lipid peroxidation, DNA damage, protein degeneration, and apoptotic damage, which have deleterious effects on normal physiological activities [16]. Moreover, high levels of ROS (e.g., H_2_O_2_) may stimulate cancer cell proliferation and apoptosis, promote cancer cell angiogenesis, and enhance cancer cell invasiveness and metastasis. Therefore, nanoparticles with antioxidant properties can not only help to alleviate the inflammatory processes and oxidative damage in the organism but also inhibit the growth of cancer cells [17]. Selenium (Se) is an essential trace element for animals and humans [18,19]. Up to now, a large body of evidence shows that selenium has an important physiological role in preventing cancer development, maintaining the normal function of the human immune system, and preventing cardiovascular diseases [20,21,22,23,24]. More importantly, Se has also been shown to exhibit excellent potential in antioxidant activity [25]. The antioxidant principle of Se is due to its heavy reliance on cofactors of antioxidant enzymes such as glutathione peroxidase, thioredoxin reductase, and superoxide dismutase. These enzymes protect cells and cell membranes from oxidative damage by catalyzing the breakdown of lipid hydroperoxides and hydrogen peroxide in cells [26]. 

In this work, we synthesized composite MnO_2_/Se nanoparticles (SM NPs) with antioxidant and catalytic functions using the strong oxidizing property of potassium permanganate and the reducing property of selenocysteine, modified with bovine serum protein (BSA) to obtain biocompatibility and colloidal stability of MnO_2_/Se-BSA nanoparticles (SMB NPs). The antioxidant properties of SMB NPs were verified by ABTS and DPPH radical scavenging studies. Moreover, the TME regulatory role of SMB NPs was verified in acid response and H_2_O_2_-catalyzed studies. MnO_2_ plays an acid-responsive and catalytic role in the tumor microenvironment to produce O_2_ by catalyzing the decomposition of hydrogen peroxide. The addition of Se, which acts as an antioxidant, helps to alleviate inflammatory processes and avoid oxidative damage. Together with the proved immunomodulatory effects of selenium [27], this study provides a feasible strategy for the design of multifunctional nanomanganese and selenium enzymes that are responsive to TME.

## 2. Results and Discussion

### 2.1. Synthesis and Characterization of SMB NPs

SM NPs were synthesized by a simple redox method in one step and modified with BSA to obtain physiologically stable SMB NPs (Figure 1a). The microscopic morphology of SMB NPs was observed by SEM. As shown in Figure 1b, the SMB NPs were spherical, uniformly distributed in size, and the average size was about 100 nm. It is worth noting that the SEM results show that the sample exhibits agglomeration, which may be caused by the drying process of the SMB suspension during sample preparation. The elemental mapping images also clearly showed the presence of Mn, Se, and O (Figure 1c–f), suggesting that the composite SMB NPs were successfully prepared.

DLS measurements were used to assess the hydrodynamic size as well as the physiological colloidal stability of the NPs (Figure 2a). It was determined that the SMB NPs did not show significant fluctuations in hydrodynamic size after 48 h storage in saline, indicating that this nanoparticle has good colloidal stability under physiological conditions. Because of the hydration layer, the hydration particle size measured by the DLS is greater than its original size (measured by SEM). However, the hydration particle size of the SMB NPs is smaller than 500 nm [28,29]. Therefore, the SMB NPs remain compatible with in vivo application. XPS analysis of Mn 2p and Se 3d was shown in Figure 2b,c. Figure 2b showed that the binding energies of Mn 2p 1/2 and Mn 2p 3/2 centered at 653.20 eV and 642.20 eV, respectively, indicating a primary oxidation state of +4 for Mn, which matched the characteristics of MnO_2_ in the literature [30]. Similarly, the binding energy of Se 3d 5/2 is centered at 58.80 eV, indicating the presence of selenite [31]. Therefore, the XPS results indicated that the SMB NPs are composed of MnO_2_ and selenite. The samples were analyzed using ICP-OES with 22.6% of Mn elements and 2.45% of Se elements after acid digestion.

### 2.2. In Vitro Degradation

The degradation performance of SMB NPs was evaluated via recording the change in the solution color and absorbance. Figure 3 showed the degradation of SMB NPs under different environments during the 24 h incubation. It was found that the decrease in absorbance and fading in color with time occurred in both saline and CBS (mimics the acidic nature of the tumor microenvironment). The degradation can be ascribed to the oxidation of Se in SMB NPs [27]. It is worth noting that the degradation rate of SMB NPs is faster in CBS than in saline, which is probably due to the pH-responsive nature of MnO_2_ (MnO_2_ converted to Mn^2+^ in an acidic environment).

### 2.3. Catalytic Response

The pH-responsive change in SMB NPs in different solutions is shown in Figure 4a. In a neutral environment (pH = 7.4), SMB NPs did not show a significant color change in the solution. However, in an acidic environment (CBS, pH = 5.4, which mimics the acidic nature of the tumor microenvironment), the SMB NPs immediately reacted with H^+^, and the MnO_2_ component was converted to Mn^2+^. Therefore, the color of the solution faded upon its contact with the CBS. The above results indicate that SMB NPs has acid response behavior. The catalytic performance of SMB NPs was evaluated by the oxygen generation capacity of MnO_2_ in the presence of H_2_O_2_. As shown in Figure 4b, H_2_O_2_ decomposed into O_2_ rapidly and generated a large number of bubbles since its contact with SMB NPs. with the bubbling lasting for at least 2 min. However, in the absence of SMB NPs, pure PBS has no obvious bubble formation. The above results demonstrate that SMB NPs can be used as a catalytic platform to effectively decompose the H_2_O_2_ in the TME simulated environment. Cancer cells typically have a high concentration of ROS (e.g., H_2_O_2_). Such a high level of ROS is an accessory to cancer cell proliferation, angiogenesis, invasiveness, and metastasis [17]. Therefore, SMB NPs provide a paradigm for the design of high-performance cancer therapeutic reagents that are responsive to the hypoxia and hydrogen peroxide overexpression nature of TME.

### 2.4. Antioxidant Activity Assessment

The pro-inflammatory response induced by cell death during tumor therapy may also affect the tumor therapy’s outcomes [32]. In recent years, nanoparticle antioxidants have become one of the emerging strategies to eliminate excess free radicals [33]. Figure 5a–d shows the free radical scavenging activities of SMB NPs with different concentrations. The scavenging of ABTS radicals is an important indicator in evaluating the overall antioxidant capacity of nanomaterials [34]. Figure 5a showed the scavenging activity of SMB NPs against ABTS^+^, and it can be seen that the scavenging activity of SMB NPs against ABTS^+^ increased with increasing SMB NPs concentration in a concentration-dependent manner, which is consistent with the results reported in the literature [35]. When the mass concentration of SMB NPs was 500 µg/mL, the ABTS^+^ scavenging ratio was as high as 67.38 ± 1.05%. The concentration-dependent ABTS^+^ scavenging activity of SMB NPs was further verified by the faded solution color (Figure 5b), where a faster color fading rate indicated a higher ABTS^+^ scavenging activity.

The ability of SMB NPs to scavenge RNS was further investigated by studying the inhibition ratio of DPPH radicals. DPPH radicals had characteristic absorption at 517 nm and their alcohol solutions were purple. The fading of the purple color and the decrease in absorbance indicated that free radicals were eliminated. From the experimental results, it is clear that antioxidant capacity increased gradually with increasing mass concentration of SMB NPs. The inhibition of DPPH by SMB NPs was as high as 78.56 ± 2.48% at an SMB NPs mass concentration of 500 µg/mL (Figure 5c). The half maximal effective concentrations (EC50) were detected as 407.58 μg/mL and 185.72 μg/mL for ABTS and DPPH, respectively (Appendix A). The color fading of the solution in Figure 5d further confirmed the excellent RNS scavenging activity of SMB NPs. In general, SMB NPs has good antioxidant properties and can maintain the balance of ROS in the body, thus enabling the body to effectively avoid inflammatory reactions analogous to the tumor therapy.

### 2.5. The In Vitro Biosafety of SMB NPs

To investigate the feasibility of SMB NPs as a multifunctional biomaterial, the biocompatibility of SMB NPs material at the cellular level was investigated and tested. The hemocompatibility of SMB NPs was investigated with an in vitro hemolysis assay. Hemocompatibility was tested using UV-Vis-NIR and then calculated. mRBCs were co-cultured with deionized water as a positive control group, and mRBCs were co-cultured with PBS as a negative control group. The hemolysis ratio of solutions after incubation with different concentrations of SMB NPs (50, 100, and 200 μg/mL) with mRBCs was less than 5%, and the results were calculated as 2.13 ± 0.70%, 0.3 ± 0.1%, and 3.13 ± 0.29%, respectively (Figure 6a). Moreover, there was no significant cell fragmentation in mRBCs treated with different concentrations of SMB NPs, which corresponded to Figure 6b, further confirming the non-destructive and non-damaging effect of SMB NPs on the structural integrity of mRBCs and the good hemo-compatibility of the SMB NPs material.

Standard cell count CCK-8 and live/dead staining were used to assess the in vitro cytocompatibility of SMB NPs. After the co-culture of SMB NPs with L929 cells at different concentrations for 24 h, the survival ratio of L929 cells was not significantly different compared with the control group (without any treatment). Even at the highest concentration, the cell viability was still higher than 90% (98.11 ± 0.42%, Figure 6c). Live/dead staining was further used to observe the abundance of green and red fluorescence to investigate the effect of SMB NPs on cell proliferation and morphology (Figure 6d). The results showed that almost all the cells were green, indicating that the nanomaterial had good cytocompatibility.

### 2.6. The In Vivo Biosafety of SMB NPs

In addition to the blood and cell safety of the material, the safety of the material in mice was evaluated. By monitoring weight change in mice, it was possible to visually determine whether SMB NPs were harmful to the organism. The mice were male with an average body weight of about 25 g and age of six to eight weeks. SMB NPs (200 μg/mL, 200 μL) was injected into KM mice via the tail vein, mice injected with 200 μL PBS were used as the control group and the body weight changes of mice were monitored over two weeks. The data showed that the mice in the experimental group showed the same trend of body weight change as those in the control group (Figure 7a). In addition, serum biochemistry (Figure 7b,c) and routine blood parameters (Figure 8a–i) of the mice were measured on days 1, 7, and 14 to assess the in vivo blood compatibility of the material. The studied serum biochemical parameters were ALT (alanine aminotransferase), urea, AST (aspartate aminotransferase), TB (total bilirubin), and CREA (creatinine). The results showed no statistically significant difference between SMB NPs injection and the control group. The main blood routine parameters included WBC (white blood cell), RBC (red blood cell), HB (hemoglobin), MCV (mean red blood cell volume), HCT (red blood cell pressure volume), MCHC (mean red blood cell hemoglobin concentration), RDW (red blood cell distribution width), PLT (platelet count), and MCH (mean red blood cell hemoglobin content). The results showed similar data to the control group, and no significant physiological abnormalities were observed. Thus, we have demonstrated that the SMB NPs have good in vivo biosafety and can be used in studies on biological aspects.

## 3. Experimental Section

### 3.1. Materials

Potassium permanganate, 2,2′-azino-bis (3-ethylbenzothiazoline-6-sulfonic acid) (ABTS), phosphate buffer solution (PBS), and citrate buffered saline (CBS, pH = 5.4, mimics the acidic nature of the tumor microenvironment) were purchased from Aladdin Reagent Co., Ltd., (Shanghai, China). Selenocysteine (98%) and 1, 1- diphenyl-2-picrylhydrazyl (DPPH) were purchased from Shanghai Yuanye Bio-Technology Co., Ltd. BSA was obtained from J&K Scientific Technology Co., Ltd., (Beijing, China). Ethanol (99.7%) was acquired from Titan Scientific Co., Ltd., (Shanghai, China). Potassium persulfate (K_2_S_2_O_8_) and hydrogen peroxide (H_2_O_2_, 99%) were purchased from Sinopharm Chemical Reagent Co., Ltd. Mouse fibroblast cells (L929) were bought from the Institute of Biochemistry and Cell Biology (Shanghai, China). Cell counting kit-8 (CCK-8) was purchased from Dojindo Laboratories (Kumamoto, Japan). Dulbecco’s Modified Eagle Medium (DMEM) was purchased from Corning Co., Ltd. (Shanghai, China). Kunming (KM) mice four to six weeks old were bought from Beijing Vital River Laboratory Animal Technology Co., Ltd. (Beijing, China).

### 3.2. Preparation of SMB NPs 

First, 20 mg of seleno-cysteine and 10 mg of potassium permanganate were weighed and dissolved in 25 mL of distilled water, and the solution was stirred on a magnetic stirrer for 12 h to fully expose the sample to the reaction. The mixed solution was then centrifuged (10,000 rpm, 5 min) and the centrifuged product (SM) was obtained by washing three times with deionized water. The SM was dispersed in 10 mL of deionized water. A total of 0.1 g of BSA was added and then placed in a cell grinder for ultra-sonication (80%, 120 min). The final product SMB NPs was obtained after centrifuging three times (12,000 rpm, 5 min).

### 3.3. Characterization of SMB NPs 

Scanning electron microscopy (SEM, VEGA 3 SBH) was used to observe the morphology of the material. After the SMB NPs was dispersed in an aqueous solution and ultrasonically dispersed as well, 10 μL of the solution was dropped onto a carrier table and dried under an infrared lamp, and then the sample table was placed into the SEM for observation and photography. The dynamic light scattering (DLS, Nano ZS 90, Malvern, UK) technique was used to determine the particle size in solution and the stability profile. Using X-ray photoelectron spectroscopy (XPS, ESCAlab 250), the element valence state of the SMB NPs sample was studied. Inductively coupled plasma spectroscopy (ICP-OES, Thermo Fisher iCAP PRO, Waltham, MA, USA) was used to analyze the content of Mn and Se in SMB NPs.

### 3.4. Degradation of SMB NPs

A total of 5 mL of SMB NPs solution (250 μg/mL) was prepared in a 10 mL centrifuge tube filled with saline or CBS. The centrifuge tube was placed in a shaker and vibrated at 120 rpm at 37 °C. The solution was removed at the specified time and the absorbance of the solution was measured with a UV-vis-NIR spectrophotometer. The color change of the solution was photographed and recorded to study the degradation of SMB. 

### 3.5. Acid Response and Catalytic Performance

First, 200 μL of SMB NPs solution (500 μg/mL) was individually prepared in PBS and CBS, and incubated in a thermostat at 37 °C for 2 h. The acid response performance of SMB NPs was assessed by observing the color change of the solution. To study catalytic performance, 200 μL of SMB NPs solution (500 μg/mL) was prepared and incubated in a thermostat at 37 °C for 2 h. Two centrifuge tubes with 2 mL of H_2_O_2_ (30%) solution in each were mixed with 200 μL of SMB NPs solution and 200 μL of PBS (control), respectively. The catalytic performance of SMB NPs was evaluated by observing bubble production.

### 3.6. ABTS Radical Scavenging Assay

A total of 2 mL of 7.4 mM ABTS solution was mixed with K_2_S_2_O_8_ solution (2.6 mM, 2 mL) and then incubated overnight at 25 °C in the dark to obtain a stable solution of ABTS radicals (ABTS^+^). Then, the solution was diluted with anhydrous ethanol so that the absorbance value at 734 nm was 0.7 ± 0.005. To study the total antioxidant capacity of SMB, 1 mL samples at different concentrations (0, 25, 50, 100, 200, and 500 μg/mL, in DI water) were mixed with 1 mL of diluted ABTS^+^ solution and placed in the dark for 30 min (n = 3). Finally, the absorbance of the solution at 734 nm was monitored using UV-vis-NIR spectroscopy (Lambda 25, PerkinElmer, Waltham, MA, USA) [36]. ABTS^+^ radical scavenging was calculated as follows:$${\mathrm{ABTS}}^{+}\;\mathrm{scavenging}\;\mathrm{activity}\;\left(\%\right)=\left(1-\frac{A_{1}-A_{2}}{A_{0}}\right)\times 100\%$$
where A_0_, A_1_, and A_2_ represent the absorbance of the control (ABTS^+^), sample (ABTS^+^ + SMB NPs solution), and blank (SMB NPs solution), respectively.

### 3.7. DPPH Radical Scavenging Assay

DPPH (1,1-diphenyl-2-picrylhydrazyl) is a stable purple radical and its solution becomes yellow after being scavenged [37]. The DPPH solution was prepared by dissolving 1 mg of DPPH in 24 mL of anhydrous ethanol in a flask which was wrapped in aluminum foil to shield the light. To study DPPH’s radical scavenging ability, 1 mL of DPPH solution, 1.5 mL of anhydrous ethanol, and 1 mL of SMB solution with different mass concentrations (0, 25, 50, 100, 200, and 500 μg/mL, in DI water) were added to the centrifuge tubes (n = 3). After mixing well, the solutions were incubated in a dark environment for 30 min. The absorbance at 517 nm was measured using a UV-Vis-NIR spectrophotometer. The DPPH radical scavenging activity was calculated as follows [38]:$$\mathrm{DPPH}\;\mathrm{radical}\;\mathrm{scavenging}\;\mathrm{activity}\;\left(\%\right)=\left(1-\frac{A_{1}-A_{2}}{A_{0}}\right)\times 100\%$$
where A_0_, A_1_, and A_2_ represent the absorbance of the control (DPPH), sample (DPPH + SMB NPs solution), and blank (SMB NPs solution), respectively.

### 3.8. In Vitro Compatibility Evaluation

The L929 cell line was chosen as a model to evaluate the in vitro biocompatibility of the composite SMB NPs. Cells were first inoculated in 96-well cell culture plates (8 × 10^3^ cells/well) and placed in a 5% CO_2_ incubator (37 °C) for incubation. After 24 h, the medium was aspirated and 100 μL of different concentrations of SMB NPs (50, 100, 200, and 500 μg/mL) were uniformly dispersed in DMEM for 24 h to verify in vitro biocompatibility (n = 3). DMEM-treated L929 cells were used as a negative control. Cell morphology was observed and photographed with a phase contrast microscope (Leica DMIL LED, Wetzlar, Germany). The effect of different concentrations of the SMB on cell viability was then determined by the Cell Counting Kit-8 (CCK-8) method: CCK-8 solution (10 μL/well) was slowly added to the treated cells, and the absorbance value at 450 nm for each well was measured using an enzyme marker after 2 h of incubation [39,40,41,42].

The in vitro hemo-compatibility of SMB NPs solution was assessed using mouse red blood cells (mRBCs) provided by Changhai Hospital (Shanghai, China) as a model. Serum and plasma were removed by centrifugation (3000 rpm, 5 min) from the whole blood of KM mice. The mRBCs were washed three times with PBS and diluted 50 times to configure the erythrocyte suspension and were then set aside. The above mRBC suspensions were mixed with different concentrations of SMB NPs solution (50, 100, 200, and 500 μg/mL), water and PBS (n = 3). The mixture was incubated in an incubator at 37 °C for 2 h and then centrifuged at 3000 rpm for 5 min. The absorbance of the supernatant of each group at 540 nm was measured by UV-Vis NIR spectrophotometer, and the hemolytic ratio was calculated [43,44,45] as follows:$$\mathrm{Hemolytic}\;\mathrm{ratio}\left(\%\right)=\frac{D_{t}-D_{\mathrm{nc}}}{D_{\mathrm{pc}}-D_{\mathrm{nc}}}\times 100\%$$
where D_t_, D_nc_, and D_pc_ represent the absorbance values of the suspension of erythrocytes treated with SMB NPs, PBS, and water, respectively.

### 3.9. In Vivo Biosafety Evaluation

To evaluate the in vivo blood compatibility of SMB NPs, routine blood, and serum biochemical tests were performed on KM mice. KM mice injected with 200 μL of PBS in the tail vein were used as the control group, and KM mice injected with 200 μL of SMB NPs solution in the tail vein were used as the experimental group (n = 3). The animal study protocol was approved by the Institutional Review Board of the First Affiliated Hospital of Naval Medical University of the People’s Liberation Army (SYXK (Shanghai, China) 2020-0033). The weight changes of KM mice in each group were measured over 14 days. The mice were executed on day 14, and blood was collected for routine blood (using a Sysmex x-800i automated hematology analyzer) and serum biochemical (using a Beckman Coulter Unicel DxC 800 automated biochemical analyzer) index evaluation.

### 3.10. Statistical Analysis

The results were expressed as the arithmetic mean ± standard deviation. Unless specified, the sample size was three (n = 3).

## 4. Conclusions

This study explored a multimodal therapeutic nanoplatform based on redox-reaction-synthesized SMB composite nanoparticles. SM particles were formed by the reaction of potassium permanganate with selenocysteine, and further modified by BSA to achieve stability under physiological conditions. The good biosafety of composite nanoparticles was verified at the L929, mRBC, and animal levels. In the SMB NPs, manganese dioxide and selenite endowed the SMB NPs with acid-responsive and catalytic, and antioxidant properties, respectively. The weak acid response, catalytic activity, and antioxidant properties of composite nanoparticles were also verified experimentally. The results showed that the hemolysis ratio was less than 5% and the cell survival ratio was as high as 95.97% after the co-culture with L929 cells at different concentrations of SMB NPs for 24 h. In addition, the biochemical parameters, main blood routine parameters, and body weight change showed no statistically significant difference between the SMB NPs-injected group and the control group. Together with the proved immune-modulatory effects of selenium, this study provides an idea to design high-performance and comprehensive therapeutic reagents that are responsive to the hypoxia, weak acidity, and hydrogen peroxide overexpression nature of TME.

## Figures and Tables

**Figure 1 molecules-28-04498-f001:**
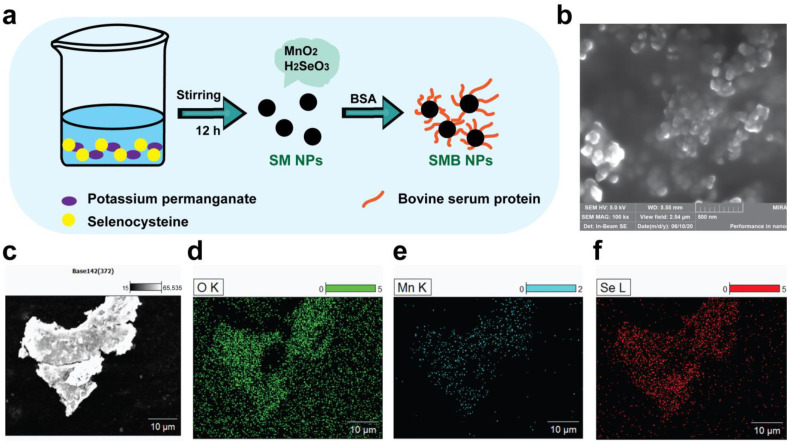
(**a**) Schematic illustration of the preparation of SMB NPs. (**b**) SEM diagram of SMB NPs. (**c**,**d**) Elemental distribution mappings of SMB NPs: (**d**) O, (**e**) Mn, and (**f**) Se.

**Figure 2 molecules-28-04498-f002:**
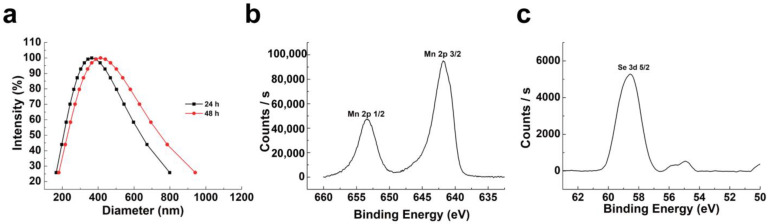
(**a**) DLS plots of SMB NPs in saline. (**b**,**c**) XPS spectra of Mn and Se.

**Figure 3 molecules-28-04498-f003:**
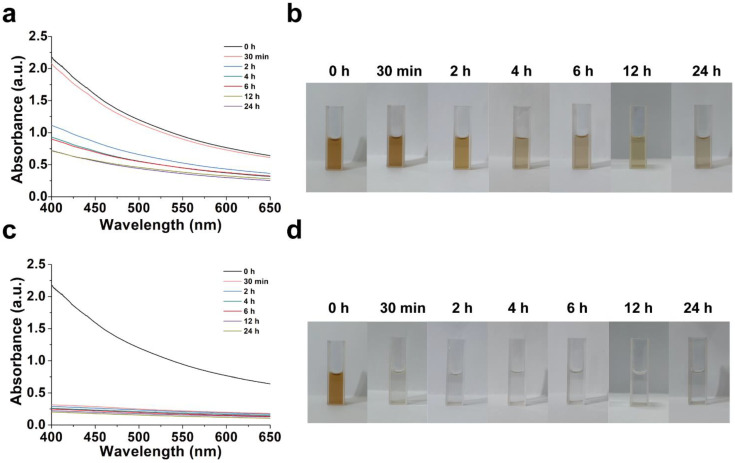
(**a**) Time-dependent UV-Vis-NIR spectra of SMB in saline. (**b**) Pictures of SMB in saline over time. (**c**) Time-dependent UV-Vis-NIR spectra of SMB in CBS. (**d**) Pictures of SMB in CBS over time.

**Figure 4 molecules-28-04498-f004:**
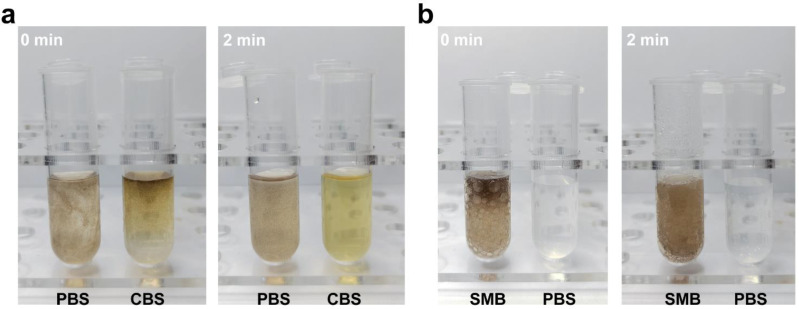
(**a**) The color change in SMB NPs placed in different buffer solutions in the first 2 min. (**b**) SMB NPs catalyzed bubble production in the first 2 min since its contact with H_2_O_2_. PBS was set as control.

**Figure 5 molecules-28-04498-f005:**
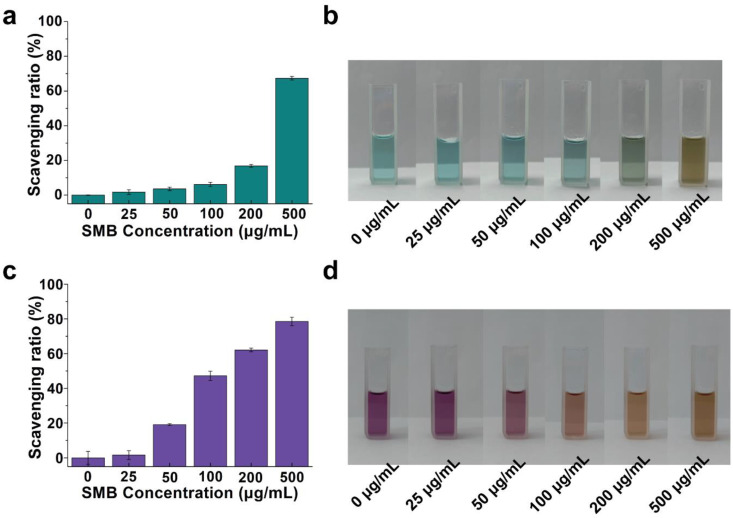
(**a**) The ABTS^+^ scavenging ratio of SMB NPs. (**b**) The color fading of ABTS^+^ solution after being treated with SMB NPs of different concentrations. (**c**) The DPPH scavenging ratio of SMB NPs. (**d**) The color fading of DPPH solution after being treated with SMB NPs of different concentrations.

**Figure 6 molecules-28-04498-f006:**
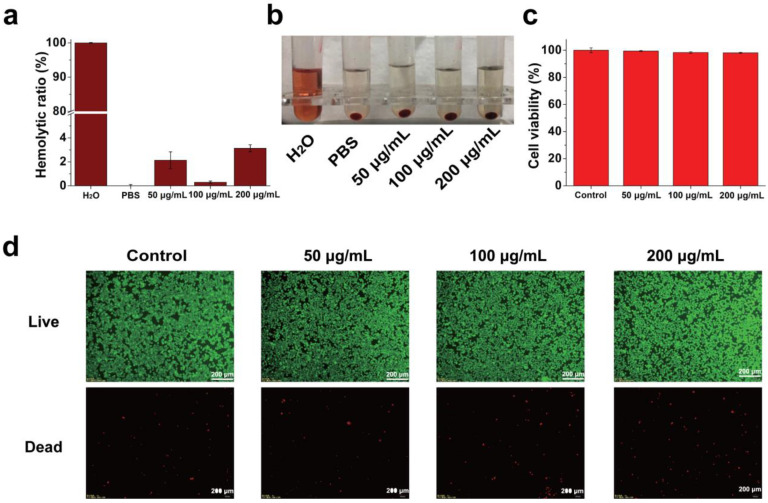
(**a**) Hemolysis ratio of mRBCs treated with water, PBS, and SMB NPs with different experimental conditions. (**b**) Images of mRBCs corresponding to (**a**). (**c**) Survival ratio of L929 cells treated with SMB NPs with different experimental conditions. (**d**) Live/dead staining images of untreated, 50, 100, and 200 μg/mL SMB NPs -treated L929 cells.

**Figure 7 molecules-28-04498-f007:**
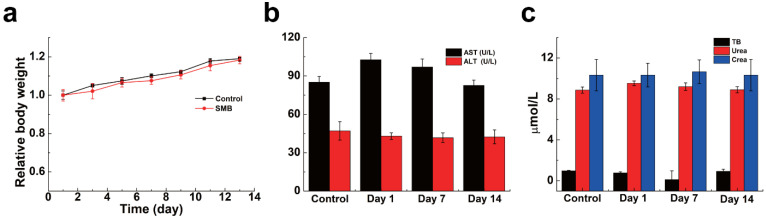
(**a**) Body weight change curves of mice treated with PBS and SMB NPs. (**b**,**c**) Serum biochemical parameters of mice at different time points after treatment with PBS and SMB NPs.

**Figure 8 molecules-28-04498-f008:**
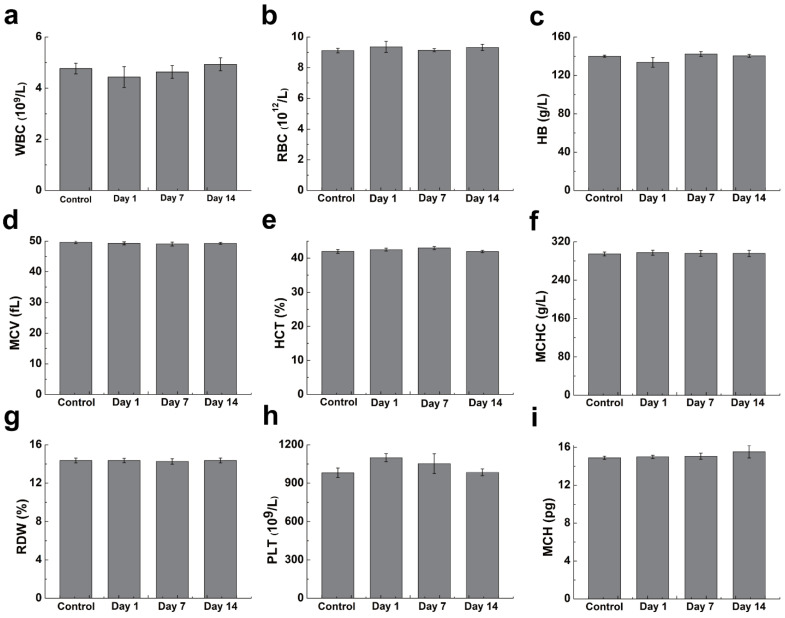
(**a**–**i**) Plots of blood parameters of mice at different time points after treatment with PBS and SMB NPs.

## Data Availability

The data that support the findings of this study are available from the corresponding author upon reasonable request.

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
