# Peer review of "Preparation of Biocompatible Manganese Selenium-Based Nanoparticles with Antioxidant and Catalytic Functions"

_molecules, 2023, doi:10.3390/molecules28114498_

Round 1
Reviewer 1 Report
The manuscript describes the preparation of MnO2/Se nanoparticles modified with bovine serum albumin with antioxidant and catalytic properties.
The authors present the results of the ROS scavenging tests and the in vitro and in vivo biosafety of the nanoparticles.
The work lacks of sufficient novelty and it is not clear which is the main contribution to the field. Indeed, it is not clear which is the connection between the antioxidant properties of the nanoparticles and their potential use in cancer. Nanoparticles in cancer therapy are generally used to kill cells, and not to restore them from an oxidative stress. There are other pathologies in which the antioxidant properties could deserve interest. Please argument about this.
Furthermore, the size and the broad size distribution of the nanoparticles, as shown by the SEM imaging and DLS curves, look to be poorly compatible with the in vivo application of the nanoparticles. The nanoparticles indeed look clustered. Discuss about this point.
Finally, it is not clear which is the benefit deriving by the combination of MnO and Se. Did the authors compare the behavior of MnO NPs versus SM NPs?
The English writing quality is good. Some typos errors have been detected into the manuscript.
Author Response
1. We thank the reviewer for the great comment. Based on the comments, we added some discussions in the revised manuscript, see page 4, lines 4-7 and below:
“Moreover, high levels of ROS (e.g. H2O2) may stimulate the cancer cell proliferation and apoptosis, promote cancer cell angiogenesis, and enhance cancer cell invasiveness and metastasis. Therefore, nanoparticles with antioxidant properties can not only help to alleviate the inflammatory processes and oxidative damage in the organism but also inhibit the growth of cancer cells [17].”
See page 12, lines 4-8 and below:
“Cancer cells typically have a high concentration of ROS (e.g., H2O2). Such a high level of ROS is an accessory to the cancer cell proliferation, angiogenesis, invasiveness and metastasis [17]. Therefore, SMB provide a paradigm for the design of high-performance cancer therapeutic reagents that are responsive to the hypoxia and hydrogen peroxide overexpression nature of TME.”
2.
We thank the reviewer for the great comment. Based on the comments, we added some discussions in the revised manuscript, see page 9, lines 11-13 and below:
“It is worth noting that the SEM results show that the sample exhibits agglomeration, which may be caused by the drying process of the SMB suspension during sample preparation.”
See page 10, lines 6-9 and below:
“Because of the hydration layer, the hydration particle size measured by the DLS is greater than its original size (measured by SEM). However, the hydration particle size of the SMB nanoparticles is smaller than that of 500 nm [35, 36]. Therefore, the SMB nanoparticles remain compatible with the in vivo application.”
3.
We thank the reviewer for the great comment. The formation of SMB is realized via a one-step redox reaction. The behavior of MnO2 alone and SM NPs were not compared in the experiments, but their respective functions were explored separately. According to this comment, we added a discussion in the revised manuscript (page 4, line 20-23). See also below:
“MnO2 plays an acid-responsive and catalytic role in the tumor microenvironment to produce O2 by catalyzing the decomposition of hydrogen peroxide. The addition of Se, which acts as an antioxidant, helps to alleviate the inflammatory processes and avoid the oxidative damage.”
Reviewer 2 Report
In this manuscript, the authors design nanoparticles of Selenite and Manganese dioxide and cover them with bovine serum protein so that they can be biocompatible. They evaluate the catalytic response of these nanoparticles as well as the antioxidant activity in non-biological systems. Subsequently, they determine the biocompatibility of these nanoparticles through a hemodialysis assay and a cell viability assay. Finally, they determine the toxicity of the SMB NPs in vivo. This manuscript is novel, however, the authors must take into account the following points for the manuscript to be considered for publication.
1. The authors use previously undescribed abbreviations throughout the abstract and introduction, which makes reading difficult.
2. In the characterization section, it remains to add the graph of Se and Mn release over time.
3. It is not clear how much Se and Mn are contained in 1g of SMB?
4. The figure corresponding to viability (5c) is incorrect, since the control must be 100% and from there see the changes with the different concentrations of SMB.
5. The images in figure 5d do not correspond to what was obtained in figure 5c,
6. Why evaluate the viability parameters at 24h and not at a longer time, how much do you estimate the half-life of the SMBs to be?
7. In the in vivo study, are the KM mice female or male and of what weight and age?
8. The authors do not mention the authorization protocol for the use of laboratory animals.
9. None of the images indicates the number of experiments that were carried out and whether it is the mean +/- the error or standard deviation.
10. The results should be discussed in detail and contrasted with what is in the literature.
Author Response
- We thank the reviewer for the great comment. Based on the comments, we have carefully checked the abstract and introduction section.
-
We thank the reviewer for the great comment. In the revised manuscript, we added the degradation performance study of SMB, which showed that the SMB is degradable. Therefore, we believe that it is not meaningful to study the elemental release given that the SMB is constantly degrading.
-
We thank the reviewer for the great comment. Based on the comments, we studied the contents of Se and Mn elements in SMB using ICP-OES. See page 6, lines 4-5 and below:
“Inductively coupled plasma spectroscopy (ICP-OES, Thermo Fisher iCAP PRO) was used to analyze the content of Mn and Se in SMB NPs.”
See page 10 , lines 14-15 and below:
“The samples were analyzed using ICP-OES, which proved that SMB have 22.6% of Mn elements and 2.45% of Se elements.”
-
We thank the reviewer for the great comment. Based on the comments, we carefully checked and calculated the data for the control group and corrected Fig. 6c.
Fig. 1 Survival ratio of L929 cells treated with SMB NPs with different experimental conditions -
We thank the reviewer for the great comment. Based on this comment, we supplemented the live-dead cell stained images in Fig. 6d. See also below:
Fig. 2 Live/dead staining images of untreated, 50, 100, and 200 μg/mL SMB NPs treated L929 cells. -
We thank the reviewer for the great comment. The in vitro degradation experiments proved that the material is degradable. Therefore, SMBs may generate ions (degradation products) due to degradation and these ions may accumulate in the culture plate and eventually lead to cell dehydration. However, this is not the case in animals because the degradation products will be metabolized without staying in the in vivo environment. Thus, we believe that a monitoring time of 24 hours for cell cytocompatibility is enough. In addition, because of the degradation of SMBs, it is not possible to measure the half-life.
See page 6 , lines 6-11 and below:
“A 5 mL of SMB solution (250 ug/mL) was prepared in a 10 mL centrifuge tube filled with saline or CBS. The centrifuge tube was placed in a shaker and vibrated at 120 rpm at 37°C. The solution was removed at the specified time and the absorbance of the solution was measured by UV-vis-NIR spectrophotometer. The color change of the solution was photographed and recorded to study the degradation of SMB.”
See page 10-11 and below:
“The degradation performance of SMB was evaluated via recording the change of the solution color and absorbance. Fig. 3 showed the degradation of SMB under different environments during the 24h incubation. It was found that the decrease in absorbance and fading in color with time occurred in both saline and CBS. The degradation can be ascribed to the oxidation of Se in SMB [27]. It is worth noting that the degradation rate of SMB is faster in CBS than saline, which is probably due to the pH-responsive nature of MnO2 (MnO2 conversed to Mn2+ in an acidic environment).”
Fig. 3 a) Time dependent UV-Vis-NIR spectra of SMB in saline. b) Pictures of SMB in saline over time. c) Time dependent UV-Vis-NIR spectra of SMB in CBS. d) Pictures of SMB in CBS over time. -
We thank the reviewer for the great comment. The KM mice used for in vivo safety experiments were added in the manuscript. See page 16, lines 4-5 and below:
“The mice were male with the average body weight of about 25g and age of 6-8 weeks.”
-
We thank the reviewer for the great comment. We added this point in the manuscript. See page 8, lines 12-14 and below:
“The animal study protocol was approved by the Institutional Review Board of the First Affiliated Hospital of Naval Medical University of the People’s Liberation Army (SYXK(Shanghai) 2020-0033).”
-
We thank the reviewer for the great comment.Based on the comments, we have added the number of samples used for the experiment in the experimental section. See page 9, lines 4-5 and below:
“The data were expressed as the mean ± standard deviation. Unless specified, the sample size was three (n = 3).”
-
We thank the reviewer for the great comment. Based on this comment, we added some analysis of experimental results and comparison with literature in the results and discussion section.
Page 10, lines 9-13:
“XPS analysis of Mn 2p and Se 3d was shown in Figs. 2b-c. Fig. 2b showed that the binding energies of Mn 2p 1/2 and Mn 2p 3/2 centered at 653.20 eV and 642.20 eV, respectively, indicating a primary oxidation state of +4 for Mn, which matched the characteristics of MnO2 in the literature [35]. Similarly, the binding energy of Se 3d 5/2 is centered at 58.80 eV, indicating the presence of selenite [36].”
Pages 12-13:
“Fig. 5a showed the scavenging activity of SMB against ABTS+, and it can be seen that the scavenging activity increased with increasing SMB concentration, which is consistent with the results reported in the literature [40].”
Page 13, lines 12-14:
“In general, SMB has good antioxidant properties and can maintain the balance of ROS in the body, thus enabling the body to effectively avoid inflammatory reactions analogous to the tumor therapy.”

Round 2
Reviewer 2 Report
The authors took into account the previous comments; however, some important points still need to be addressed in order for it to be considered for publication:
1. After reviewing the introduction again, I was left with the doubt about the use that authors propose for these Se/Mn nanoparticles. Since in line 27 of the manuscript is mentioned that tumor cells have high amounts of GSH, which is one of the main antioxidants, and later in the results is mentioned other over expressions such as SOD, which is also an antioxidant enzyme, so putting it in that context it is not clear why making nanoparticles with more antioxidants. Rather, I believe that Se/Mn nanoparticles are feasible to aid cancer treatment, that is, chemo and radiotherapy, which, if it is known that they induce inflammation and oxidative stress, I believe that this point should be made very clear in the introduction and results, otherwise the proposal would be contradictory.
2. In the abstract it is necessary to indicate what SMB NPs means.
3. In line 114 what is the meaning of CBS.
4. In the antioxidant capacity section of the nanoparticles, the EC50 value must be entered for each of the tests.
5. Homogenize the use of SMB or SMB NPs.
6. In figure 7b it is not observed that it is measured on the 'y' axis